# Software-Defined Networking Orchestration for Interoperable Key Management of Quantum Key Distribution Networks

**DOI:** 10.3390/e25060943

**Published:** 2023-06-15

**Authors:** Dong-Hi Sim, Jongyoon Shin, Min Hyung Kim

**Affiliations:** SK Telecom, Seoul 04539, Republic of Korea; jongyoon.shin@sk.com (J.S.);

**Keywords:** QKD, SDN, SDN orchestrator, SDN controller, interoperable KMS, interworking

## Abstract

This paper demonstrates the use of software-defined networking (SDN) orchestration to integrate regionally separated networks in which different network parts use incompatible key management systems (KMSs) managed by different SDN controllers to ensure end-to-end QKD service provisioning to deliver the QKD keys between geographically different QKD networks. The study focuses on scenarios in which different parts of the network are managed separately by different SDN controllers, requiring an SDN orchestrator to coordinate and manage these controllers. In practical network deployments, operators often utilize multiple vendors for their network equipment. This practice also enables the expansion of the QKD network’s coverage by interconnecting various QKD networks equipped with devices from different vendors. However, as coordinating different parts of the QKD network is a complex task, this paper proposes the implementation of an SDN orchestrator which acts as a central entity to manage multiple SDN controllers, ensuring end-to-end QKD service provisioning to address this challenge. For instance, when there are multiple border nodes to interconnect different networks, the SDN orchestrator calculates the path in advance for the end-to-end delivery of keys between initiating and target applications belonging to different networks. This path selection requires the SDN orchestrator to gather information from each SDN controller managing the respective parts of the QKD network. This work shows the practical implementation of SDN orchestration for interoperable KMS in commercial QKD networks in South Korea. By employing an SDN orchestrator, it becomes possible to coordinate multiple SDN controllers and ensure the efficient and secure delivery of QKD keys between different QKD networks with varying vendor equipment.

## 1. Introduction

The use of secure network applications such as multimedia, big data, AI, and autonomous vehicles and the proliferation of 5G technology have resulted in an increase in security threats. Presently, the distribution of secret keys used for high-speed encryption communications is mostly dependent on public-key cryptography systems which rely on computational complexity. However, the advancement of computing power and the development of quantum computing have intensified the potential for hacking threats. As a result, the use of quantum encrypted communication with quantum key distribution (QKD) [1], which offers complete protection for crucial data, has become increasingly important, necessitating effective management and the construction of infrastructure.

Software-defined networking (SDN) is a potential solution for managing QKD-enabled networks [1,2,3,4,5]. SDN is a network architecture that separates the network control plane from the data plane, allowing for centralized management and control of the network [3,4]. This can provide several benefits for QKD-enabled networks, including the ability to dynamically allocate network resources, manage network security policies, and optimize network performance. By utilizing SDN, QKD-enabled networks can be managed more effectively and efficiently.

In [3,4], a QKD network enabled by SDN by elaborating on its overall architecture, related interfaces, and protocols is presented. Further, the previous works in [2,5] demonstrated SDN services utilizing QKD technologies that were fully integrated with standard telecommunication networks. However, there is a lack of studies addressing orchestration with SDN functionality between different QKD networks to deliver the keys. This can be particularly challenging in large-scale communication networks where the QKD network is made up of parts from several different vendors. This challenge must also be solved when connecting the QKD networks of different operators.

The objective of this work is to demonstrate, for the first time, SDN orchestration for interoperable key management systems (KMSs) to deliver QKD keys to cryptographic applications in which different parts of a network use KMSs that are incompatible as the network is composed of different QKD devices and KMSs from different vendors. The use case is the delivery of QKD keys when different parts of a network use KMSs that are managed separately by different SDN controllers which require an SDN orchestrator to manage. It is assumed to have common trusted nodes, which are border nodes, to interconnect different parts of networks. This is common practice, and it is sometimes mandatory for operators to have multiple vendors for deploying network equipment in their own network. This practice also facilitates the extension of the coverage of the QKD network, allowing for the interconnection of different QKD networks equipped with QKD devices from various vendors. Given that the development of an extensive and global QKD network is still in an early stage, it is crucial to interconnect different QKD networks equipped with various QKD devices. This facilitates the expansion of QKD network coverage for key delivery, but it requires the operator to coordinate the SDN controllers that manage each QKD network individually. As coordinating different parts of a QKD network is extremely complicated, an SDN orchestrator is implemented to manage multiple SDN controllers to ensure end-to-end QKD service provisioning between different parts of a QKD network with different vendors. For example, if there are multiple common trusted nodes to interconnect different parts of systems, a path must be calculated in advance for an end-to-end key delivery from an initiating application to the target application belonging to different parts of a network, meaning a common trusted node must be selected in advance. The SDN orchestrator is designed and implemented to perform this role with the information from each SDN controller from each part of the QKD network.

The SDN orchestrator facilitates the integration and control of different networks composed of various QKD vendors, providing user portals for configuring services and an intelligent monitoring system for increasing operational efficiency and flexible network scalability. Additionally, the SDN orchestrator provides network information, such as topology, nodes, links, inventory, services, performance, events, and cryptographic key status, as well as offering automatic network monitoring and network management system (NMS) functions such as troubleshooting and incident management.

## 2. Integration of Quantum Nodes and Systems in the Current Network

### 2.1. SDN Controller

To optimize the transmission of quantum and classical signals over a network and to manage the key relay for longer distances, it is necessary to integrate QKD systems with network control at both the physical and logical levels. This can be achieved by describing the capabilities of QKD devices to the network controller using the YANG modeling language [6,7,8,9,10]. The purpose of this information model is to simplify the management of QKD resources by implementing an abstraction layer in YANG, which will allow for the efficient creation and usage of QKD-derived keys through the use of a central element, the SDN controller, which has a global view of the network.

An SDN-enabled QKD node (SD-QKD node) is a collection of one or multiple QKD modules that connect to an SDN controller using standard protocols, allowing it to be remotely and dynamically configured by the controller to create, remove, or update key associations between remote locations, either through a quantum channel or multi-hop routing with trusted nodes.

### 2.2. SDN Orchestrator

A typical use case for using an SDN orchestrator in building a QKD network involves efficiently managing two different SDN controllers responsible for managing an optical transport network (OTN) and the QKD network, respectively [11], when network operators choose to separate quantum channels in a QKD network and classical data channels in an OTN, with each network controlled by different SDN controllers. When QKD-derived keys are to be used in secure application entities in an OTN, network operators need to know which QKD nodes in the QKD network can supply those keys. Adopting an SDN orchestrator that can control both the QKD and OTN networks via each SDN controller achieves end-to-end service provisioning of QKD-derived key generation and use. The SDN orchestrator matches the addresses of the QKD nodes in the QKD network and secure application entities in the OTN to supply the keys. Additionally, an SDN orchestrator interfaced with the SDN controller of the QKD network can be used to manage and configure the QKD network, reducing the burden of integrating the QKD network into existing communication networks. The information model for the interface between the SDN orchestrator and the SDN controller of the QKD network is presented to simplify management and configuration through the northbound interfaces of the SDN controllers of the QKD networks. This functionality is defined in the ETSI GS QKD 018 standard [11].

## 3. Interworking of Two Different QKD Networks

The main objective of the key management system in a QKD network is to provide QKD keys to cryptographic applications in which cryptographic key-consuming applications at each node are referred to as secure application entities (SAEs), and the key management software they connect to are referred to as key management entities (KMEs). The key management system of the network is responsible for securely distributing keys between the KMEs, allowing key requests from the initiator and target SAEs to be fulfilled. APIs have been defined in ETSI GS QKD 014 [12] and ETSI GS QKD 004 [13] to specify methods and data formats for delivering keys from a KME to an SAE. However, if different parts of a network use incompatible or separately managed key management systems, a standardized interface is needed for KMEs to transfer keys horizontally between different parts of the network. This standard interface has been addressed in ETSI GS QKD 020 [14].

The objective of this paper is to demonstrate the practical implementation of an SDN orchestrator that efficiently manages SDN controllers governing each QKD network, with the aim of enabling the transmission of QKD keys between two networks that employ distinct vendors.

## 4. Case Study of Interworking of QKD Networks and the Role of SDN Orchestrator

This study pertains to the integration of operational and control mechanisms aimed at ensuring the efficient operation and maintenance of heterogeneous QKD equipment and transmission equipment. This research is based on a practical case study conducted within the Korea Research Environment Network (KOREN) in South Korea. The objective of this research is to not only manage individual SDN controllers for QKD equipment and transmission equipment but also to enable the exchange of keys between different domains, i.e., networks, that are associated with disparate QKD equipment.

### 4.1. Korea Research Environment Network (KOREN) Overview

KOREN is a national research and education network that provides advanced network services for the research and education community in South Korea. KOREN connects researchers, educators, and students from universities, research institutions, and government agencies to each other and to international research networks and resources. It provides high-speed internet connectivity, cloud services, and other advanced network services to support research, education, and collaboration. Additionally, KOREN also operates a testbed network for experimentation and innovation in advanced networking technologies.

KOREN is a research backbone network that connects 10 regional access points across South Korea at speeds ranging from 10 Gbps to 260 Gbps. Subscriber circuits have been provided at speeds ranging from 1 G/10 G to a maximum of 25 G since 2020. The KOREN network in South Korea is also connected directly to international networks, such as those in Hong Kong and Singapore, with a 100 Gbps bandwidth and supports international collaborative research with the USA and Europe.

### 4.2. Implementation Results of SDN Controller and SDN Orchestrator for KOREN QKD Networks

#### 4.2.1. General Configuration and Overview of the Use Case

The specific use case implemented in KOREN is as follows: KOREN exists solely as a separate and independent provider of QKD keys, while the actual two QKD networks generating the keys are operated by separate QKD network operators. The two operators of the QKD networks operate in disparate geographic locations. KOREN endeavors to interconnect its network infrastructures, with the aim of facilitating a more extensive dissemination of QKD keys in the wider region. In other words, in the case in which KOREN utilizes the networks of two different QKD network operators in adjacent regions where the two QKD network operators’ networks are connected, it is necessary to coordinate the transfer of QKD keys.

To implement this, each QKD network operator installed their own QKD SDN controller (QKDNC) responsible for managing their QKD network, and the project was carried out in the form of having a QKD SDN orchestrator (QKDNO) responsible for managing the two QKDNCs in KOREN. KOREN is a service provider that provides actual QKD keys, so this can be also viewed as a QKD provider utilizing different QKD vendors to integrate regionally separated networks from a different perspective.

The network topology of the testbed implemented between 2020 and 2022 is depicted in Figure 1. The details of implementation of the testbed deployed by SK Telecom are provided in Table 1.

The QKD network under consideration comprises multiple nodes, some of which belong to different QKD network operators. Specifically, the nodes highlighted in orange and light blue correspond to QKD nodes associated with distinct operators. Border nodes, which interconnect different QKD network operators, are also present, with nodes Pangyo and nodes Daejeon being examples of such nodes. Specifically, the use of border nodes allows for the establishment of secure connections between different QKD network operators, thereby expanding the range of available network resources and enhancing network resilience. Such interconnections can be particularly beneficial in scenarios in which a single QKD network operator may not be able to provide sufficient coverage or when interconnecting networks can lead to increased security and reliability. A QKD node configuration and the links between nodes such as the QKD link and KMS link and an encryptor link between QKD nodes are depicted Figure 2. This is a typical QKD node configuration that links nodes, such as the links between Seoul and Suwon in Figure 1. Figure 3 shows a photo of a QKD node in SK Telecom in Seoul as an example. The quantum bit error rate (QBER) and the key rate of each link which was actually measured in the testbed are shown in Figure 4.

A use case tested for the interoperable key management configuration with a border node between two different QKD networks with each different vendor is depicted in Figure 5, which is described in clause 3. An example of this configuration corresponds to the links among Seoul–Pangyo–Icheon in Figure 1, where Pangyo is a border node. There are no QKD link or encryptor link in the border node, which contains only two KMSs which each belong to each different QKD network operator and are interconnected to deliver the keys between the two different QKD networks.

#### 4.2.2. Implementation of QSDNO

The QKDNO of KOREN holds information on two interconnected networks, each with its own vendor, in advance. When a request for interconnection between the two networks is received, it decides the route for processing the received request using the information of the interconnected networks it holds in advance and delivers the route to the QKDNC of each network domain. Afterward, the QKDNC of each network domain carries out the internal processing for the specific route that requires processing in the segment of the network domain it governs.

For the end-to-end QKD-derived key delivery, the QKDNO can request path calculations from the path calculation engine (PCE). The PCE repeatedly provides a list of candidate paths that satisfy the constraints received from the QKDNO until the QKDNO finally chooses a path and decides to deploy it for the QKD service link in the QKD network. When a QKD service provider such as KOREN deploys different KMSs for the different parts of a network, it needs to know where common trusted nodes are located with APIs to allow the KMSs to interoperate to pass keys between the two different QKD networks. If there are multiple common trusted nodes to interconnect different QKD networks, a path needs to be calculated in advance for an end-to-end key delivery from the initiating application to the target application; these each belong to different QKD networks, which means a common trusted node must be selected in advance. The QKDNO can perform this role with the information from each QKDNC of each QKD network. For the QKDNO to coordinate between network parts, an interface between the QKDNO and the QKDNC needs to be defined. This interface describes the flow of information between both entities, with the QKDNC performing as a server and the QKDNO operating as a client. The QKDNO can orchestrate QKD networks through this interface in terms of network configuration and topology, management policy, and performance management.

Figure 6 illustrates the process of transmitting keys between different QKD networks. The QKDNO is connected to each QKDNC that manages a different QKD network, and this connection occurs through the SDN orchestrator interface. When an application request for key delivery between two applications, which are indicated as SAE_ID1 and SAE_ID2 in Figure 2, belonging to different QKD networks is received, a service link connecting the two applications in the two networks is calculated by QKDNO, and a path is designed to pass through a specific border node. Subsequently, each service link belonging to each network is established by the QKDNC of each network, which are represented as “Service Link for QKD network A” and “Service Link for QKD network B” in different colors in Figure 2, respectively, and at this border node, a separate API is used to deliver the key, which was discussed in clause 3. An example of this is standardized in ETSI GS QKD 020 [14].

To create a service link between two applications, SAE_ID1 and SAE_ID2, which exist in two different networks, the QKDNO calculates the path that the service link must traverse through the path computation engine. If the service link needs to pass through different network domains to be realized, an ID corresponding to the multi-segment, “ms_app_id”, is set to achieve the service link. The application must include relevant information, such as “server_ms_info” and “client_ms_info” in Table 2, to perform the procedure of transmitting the keys between the two networks, expressed in the figure as “extkey API”, and this paper refers to ETSI GS QKD 020 to implement it. Subsequently, the QKDNC overseeing each network domain copies this “ms_app_id” to set up the service link for each network segment.

#### 4.2.3. Data Model of SDN Orchestrator for Interoperable KMS

When two networks separated by a specific QKD vendor are interconnected, a new data model is required. The parameters of a QKD application to support an interoperable KMS are described in Table 2. The data model used here is based on the parameters specified in ETSI GS QKD 015 [6]. As per the current ETSI GS QKD 015 data model, only “client” and “internal” exist; thus, it is not possible to address the use cases for inter-device and inter-network integration. Therefore, a new data model called “multi-segment” has been defined to support multi-vendor and inter-network integration, as described in Table 2. In other words, a new application type called “multi segment” in YANG for interworking cases is defined, and this application type is only set when interworking is necessary.

If we refer to the two SD-QKD nodes of “service link A for network A” in Figure 6 as node A and node B, respectively, the YANG data model between the two nodes can be represented in Figure 7 as an example.

To invoke the key delivery process for integrating heterogeneous KMSs, the necessary information is passed as parameters in “rpc:app-registration-ms”, and the app_type is defined as MS (multi-segment) to provide MS information. In the figure, “extkeys” refers to the transmission of keys at the border node between QKD network A and QKD network B for the interoperable KMS described in clause 3.

#### 4.2.4. General Procedure

Figure 8 shows the exchange of messages between the user application and the network operator with QKDNC as well as QKDNO. General users refer to those who register on the user portal and apply to test and use the QKD service link on the KOREN test network. Network administrators manage the KOREN test network and are responsible for generating and managing service links requested by general users.

When an interconnection request is received, the QKDNO of KOREN leverages its pre-existing knowledge of the two networks, each with its own vendor, to determine the processing route for the request, which is then transmitted to the QKDNC of each network domain for further processing of the specific route within their respective segments. It is feasible to generate a service and alter its path by transmitting a control remote procedure call (RPC) from the QKDNO to the QKDNC.

This paper primarily focuses on the roles of the QKDNO and QKDNC in the process of transmitting keys between different QKD networks, as well as the development of related service flows and YANG models. Therefore, it only provides an explanation of this process with an RPC for the connectivity services between the QKDNO and QKDNC as follows:Retrieve topology from each QKDNC: QKDNO retrieves the network topology information only relevant for the interoperable KMS from each QKDNC.Application registration from QKDNO and QKDNC:When the QKDNO receives an application request from a network administrator, it uses a path computation engine to design a path that can accommodate the request, taking into account the need for interconnection between different QKD networks if required.Create service link:When an application requests a service, the QKDNO or QKDNC assigns a service link ID for the application and creates a combination of the application and service link. Since the node is specified, information can be sent to the KMS, but there is currently no path information.Create candidate paths per service link:Creates candidate paths per service link for an application. If necessary, continuous update requests can be made through a separate update RPC before selecting the final candidate path.Deploy candidate paths per service link:QKDNO-selected paths are deployed per service link for the application.

## 5. Conclusions

The demonstrations in this paper present a clear step forward in enabling the integration of different QKD networks to deliver the keys between them via a QKDNO to manage geographically separated networks in which the different networks use different KMSs managed by each QKDNC. It is the first QKDNO to interconnect different QKD networks managed by different QKDNCs for the interoperable KMSs in the commercial network. This implementation can enable the extension of QKD network coverage by interconnecting various QKD networks equipped with different devices from different vendors. This paper presents the commercial and practical implementations of an QKDNO which acts as a central entity to manage multiple QKDNCs, ensuring end-to-end QKD service provisioning in South Korea, which is realized by SK Telecom.

## Figures and Tables

**Figure 1 entropy-25-00943-f001:**
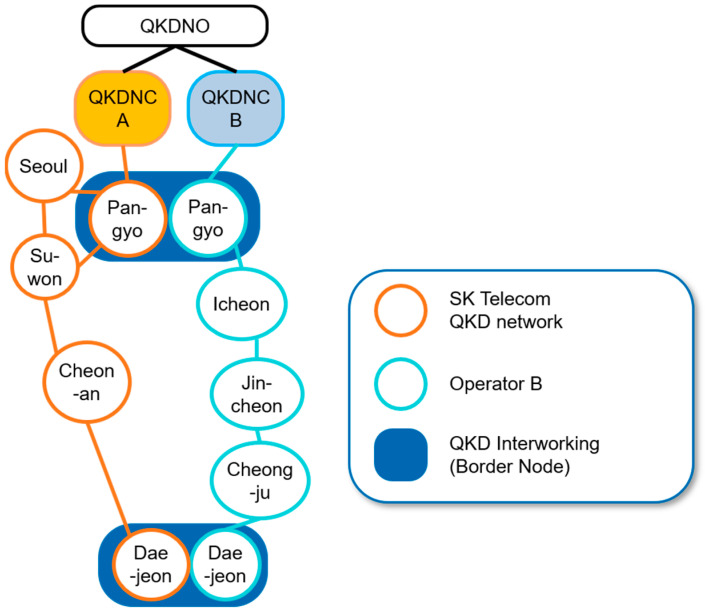
Network topology of the testbed.

**Figure 2 entropy-25-00943-f002:**
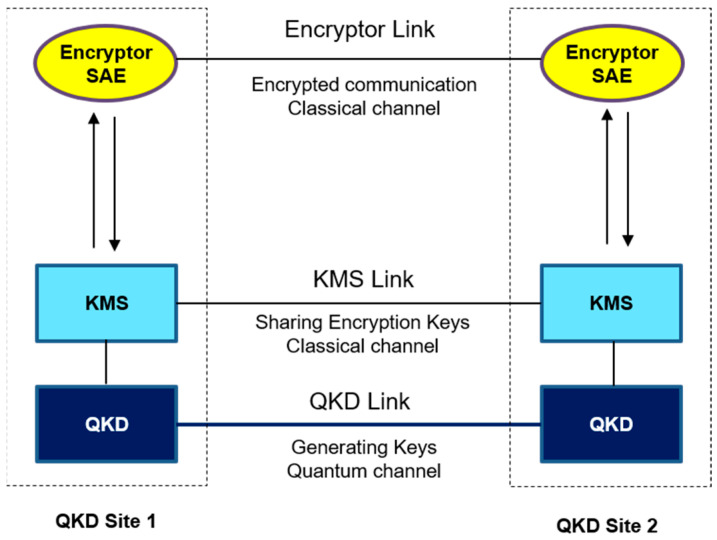
QKD node configuration and QKD/KMS/encryptor links.

**Figure 3 entropy-25-00943-f003:**
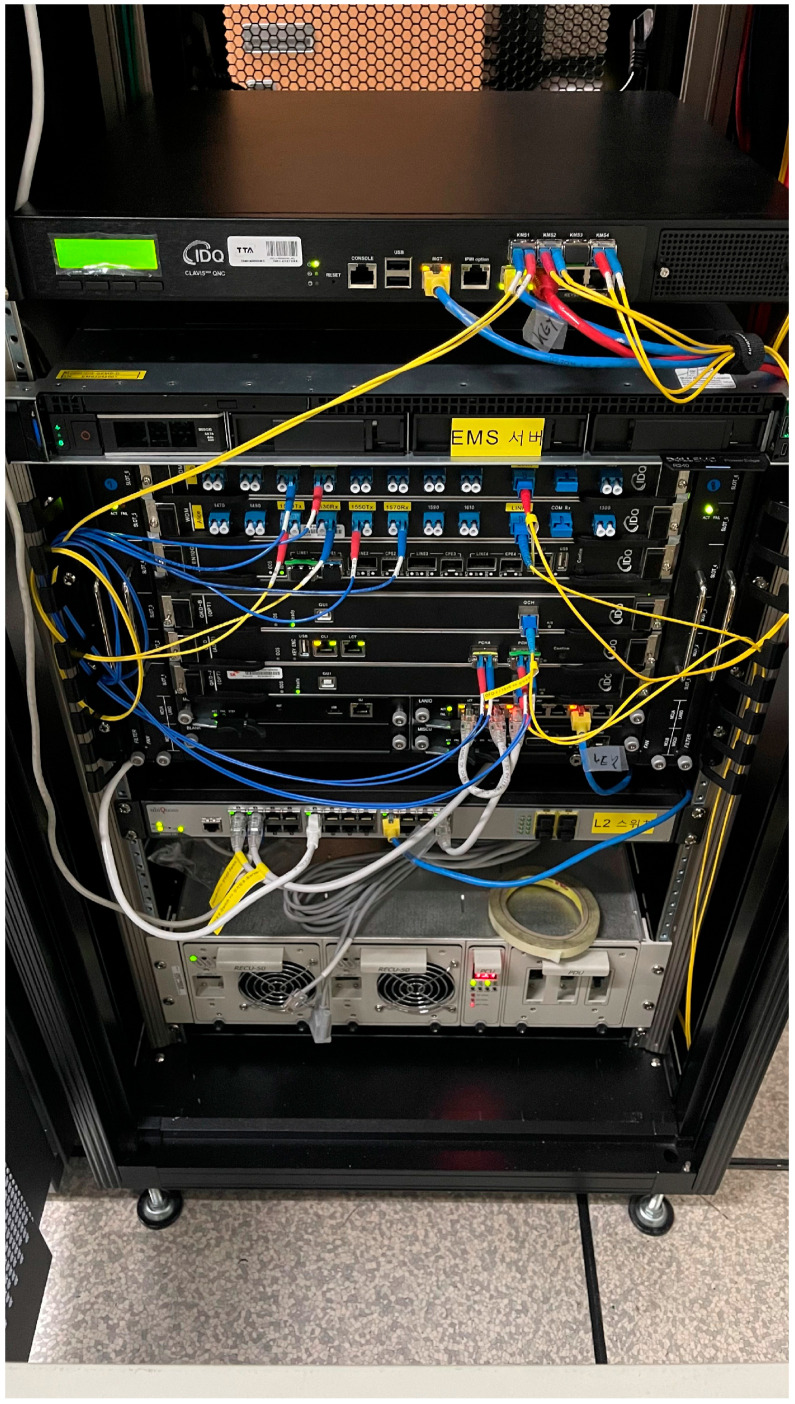
QKD node in SK Telecom in Seoul.

**Figure 4 entropy-25-00943-f004:**
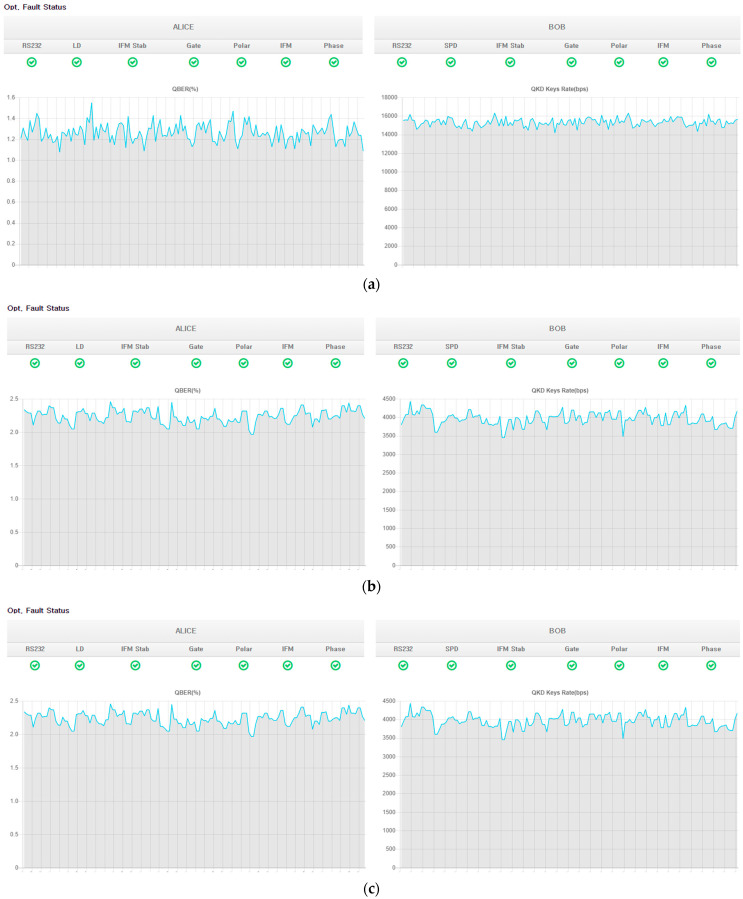
QBER (left side) and key rate (right side) of each link in the testbed. (**a**) Link between Seoul and Pangyo. (**b**) Link between Seoul and Suwon. (**c**) Link between Suwon and Pangyo.

**Figure 5 entropy-25-00943-f005:**
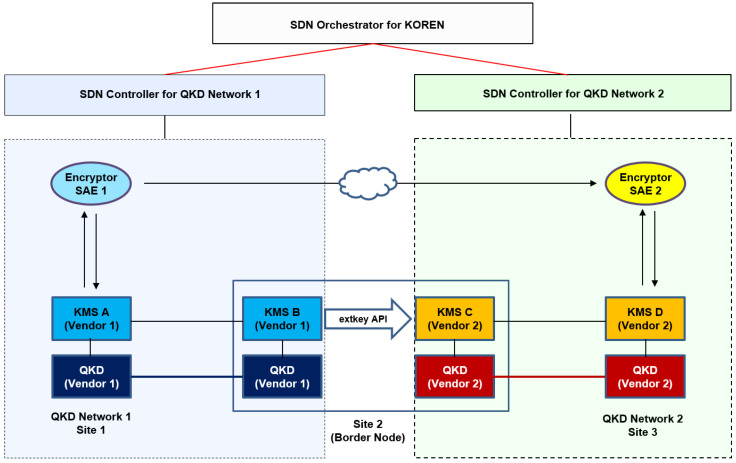
A use case for the interoperable key management with a border node, i.e., Site 2, between two different QKD networks with each different vendor. Examples of Site 2 are border nodes in Pangyo and Daejeon in Figure 1.

**Figure 6 entropy-25-00943-f006:**
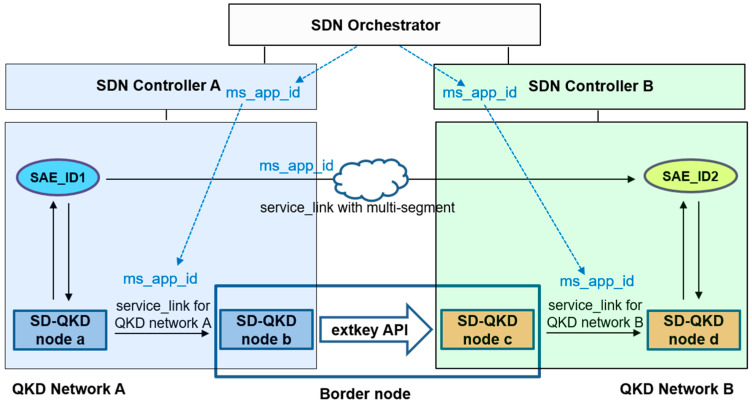
An example of the creation of the service_link with the multi-segment for the use case of delivering the key between two QKD networks via the border node is depicted.

**Figure 7 entropy-25-00943-f007:**
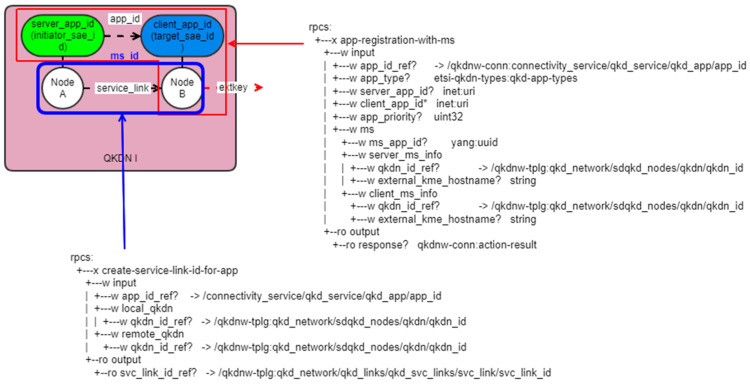
An example of the YANG data model within one QKD network for an interoperable KMS.

**Figure 8 entropy-25-00943-f008:**
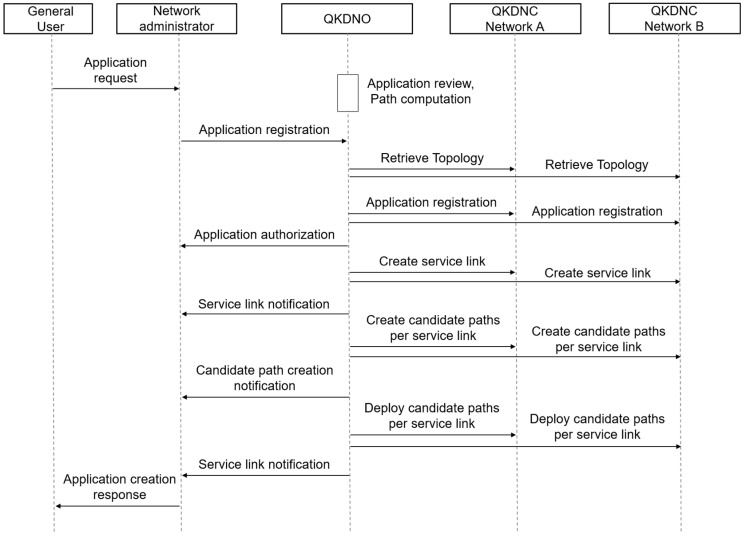
Sequence diagram of application registrations for interworking with RPC for the connectivity services.

**Table 1 entropy-25-00943-t001:** Details of implementation of testbed implemented by SK Telecom, as described in Figure 1.

Section	Distance	Protocol
Seoul–Pangyo	40 km	BB84 with decoy state
Seoul–Suwon	66 km	BB84 with decoy state
Suwon–Pangyo	39 km	BB84 with decoy state
Suwon–Cheonan	82 km	BB84 with decoy state
Cheonan–Daejeon	100 km	BB84 with decoy state

**Table 2 entropy-25-00943-t002:** Parameters of a QKD application.

Name	Type	Details	Description
app_id	ietf-yang-types: uuid	None	This value uniquely identifies a QKD application consisting of a set of entities that are allowed to receive keys shared with each other from the SD-QKD nodes they connect to.
app_status	etsi-qkdn-types: app-status-types	None	Status of the application
app_type	etsi-qkdn-types: qkd-app-types	None	Type of the registered application. These values, defined within the types module, can be client (if an external application is requesting keys) or internal (if the application is defined to maintain the QKD, e.g., multi-hop, authentication, or other encryption operations). In case of delivering a key across multiple QKDNs, the type is multi-segment.
qkd_app/app_priority	uint32	None	Priority of the association/application. This might be defined by the user, but it is usually handled by a network administrator.
server_app_id	inet:URI	None	ID that identifies the entity that initiated the creation of the QKD application to receive keys shared with one or more specified target entities identified by client_app_id. It is a client in the interface to the SD-QKD node, and the name server_app_id reflects that it requested the QKD application to be initiated.
client_app_id	leaf-list: inet:URI	None	List of IDs that identify the one or more entities that are allowed to receive keys from SD-QKD node(s) under the QKD application in addition to the initiating entity identified by server_app_id.
ms/ms_app_id	ietf-yang-types: uuid	None	This ID is used for an “end-to-end” multi-segment application to differentiate from the local app ID in each network domain to support the use of an interoperable KMS on a global scale.
ms/server_ms_info /external_kme_hostname	String	None	Hostname or IP address and optionally the port of the external KMS from which the keys are delivered or relayed.
ms/server_ms_info /qkdn_id/	ietf_yang_types:uuid	None	Unique ID of the local SD-QKD node which provides QKD keys to the remote application or external KMS. While unknown, the local SD-QKD will not be able to provide keys to the local application.
ms/client_ms_info/ external_kme_hostname	String	None	Hostname or IP address and optionally the port of the external KMS to which the keys are delivered or relayed.
ms/ client_ms_info/qkdn_id/	ietf_yang_types:uuid	None	Unique ID of the remote SD-QKD node which provides QKD keys to the remote application or external KMS. While unknown, the local SD-QKD will not be able to provide keys to the local application.
app_qos	container	None	Requested quality of service.
qos/ max_bandwidth	uint32	None	Maximum bandwidth (in bits per second) allowed for this specific application. Exceeding this value will raise an error from the local key store to the application. This value might be internally configured (or by an admin) with a default value.
qos/ min_bandwidth	uint32	None	This value is an optional QoS parameter that enables a minimum key rate (in bits per second) for the application.
qos/ jitter	uint32	None	This value allows for the specification of the maximum jitter (in milliseconds) provided by the key delivery API for applications requiring fast rekeying. This value can be coordinated with the other QoS to provide a wide enough QoS definition.
qos/ ttl	uint32	None	This value is used to specify the maximum time (in seconds) that a key could be kept in the key store for a given application without being used.
backing_qkdn_id	leaf-list: ietf-yang-types: uuid	None	Unique ID of the key association link which provides QKD keys to these applications.

## Data Availability

The data presented in this study are available on request from the corresponding author. The data are not publicly available due to privacy.

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
