# Peer review of "Software-Defined Networking Orchestration for Interoperable Key Management of Quantum Key Distribution Networks"

_entropy, 2023, doi:10.3390/e25060943_

Round 1

Reviewer 1 Report

The manuscript proposes a procedure for integrating a multinode quantum key distribution (QKD) network into the current telecommunications infrastructure through a software defined network (SDN) orchestrator that can mediate between QKD networks operated by independent QKD vendors. 

There are three primary challenges to incorporating QKD systems into existing  optical communication platforms: limited transmission distance due to losses, sensitivity of quantum signals to noise, and incompatibility of QKD devices to standard network hardware and software. An SDN provides an architecture for enabling the QKD technology to be integrated into network through an SDN controller (SDNC) that can manage resources globally.

The paper discusses a case study of implementing an SDN orchestrator (SDNO), which allows to coordinate the transfer of QKD keys when being relayed across two QKD networks in Korea. They describe in some detail how an SDNO is employed between the SDNCs of two separate QKD networks, and the use case shows how a service link is established between one chosen node in each QKD network.

I think solving the problem of how to integrated QKD networks into the current telecom infrastructure is an important one for the promotion and development of secure quantum technologies. I am unfamiliar with the YANG data model but my impression is that it provides a conceptual methodology for the orchestration procedure. While the precise details may be interesting, I do not know how much evidence it provide to the actual feasiblity of the concept. From how the case use is described, it is unclear to me if there is an actual proof-of-principle demonstration that it works. If that were the case, this makes the manuscript a lot more appealing  so more details of this is invaluable. If the paper is only discussing the conceptual model, I feel this might not be sufficient. Until it can be clarified if there is an actual SDNO implementation with a transmitted key was achieved, I am unable to recommend it for publication.  

Author Response

Response to the reviewer regarding the paper submitted for the special issue:

Dear Sir/Madam,

Thank you for your time and efforts to review the paper.

The reviewer kindly requested the details of the actual implementation.

As the use case described in the paper was practically implemented between 2020 and 2022 in the testbed in South Korea. It is now clearly indicated in clause 4 and the details of the implementation are given in particular in clause 4.2.

Please also note that the paper attached has been heavily revised with more clear text, more figures, and tables showing the results in the live testbed. The main contribution of this paper is the use of Software-Defined Networking (SDN) orchestration to integrate regionally separated networks where different network parts use incompatible key management systems (KMS) managed by different SDN controllers to ensure end-to-end QKD service provisioning to deliver the QKD keys between geographically different QKD networks. The real data model and the procedure deployed in the SDN orchestrator are also described in clause 4 with modified texts and figures.

The authors believe that the comments from the reviewer helped to enhance the quality of the paper. The authors assert that the reviewer's comments have significantly contributed to the improvement of the paper's overall quality. The authors hope this revision clarifies the original intention and answers the questions raised by the reviewer.

Sincerely,

The authors of the aforementioned paper addressed in the heading of this cover letter

Reviewer 2 Report

This paper studies how to integrate QKD devices into the current optical communication infrastructure. A software-defined networking orchestration for QKD networks (QSDNO) is presented. Although the topic is quite interesting and meaningful, there are some serious problems.

1.     It is hard to distinguish some important information from ABSTRACT. For example, what is the innovation of the manuscript? How has the problem been resolved?

2.     Some researchers have proposed some SDN-based QKD network solutions. While those related works are not referred to at all in this manuscript, not to mention the analysis of the advantages/ disadvantages of those works.

3.     It should be clarified whether the proposed QSDNO is tested on a practical QKD network integrated into the current optical communication infrastructure. In that case, it is necessary to introduce the topology of the QKD network, including the scale of the optical communication infrastructure, the number of QKD devices, and so on.

4.     It is necessary to explain the difficulties of the communication between nodes from two different QKD networks. The specific method to overcome these difficulties should also be supplemented.

5.     The experimental results and analysis should be provided to help readers to evaluate the performance of the proposed scheme.

6.     The Section of CONCLUSION is missing.

7.     Figures 2 needs to be reorganized to make it more understandable.

Author Response

Dear Sir/Madam,

Thank you for your time and efforts to review the paper.

We have taken the time to individually address and respond to each of your comments and questions as follows:

1.     It is hard to distinguish some important information from ABSTRACT. For example, what is the innovation of the manuscript? How has the problem been resolved?

Please note that the abstract has been rewritten to emphasize the main contribution of the paper attached.

2.     Some researchers have proposed some SDN-based QKD network solutions. While those related works are not referred to at all in this manuscript, not to mention the analysis of the advantages/ disadvantages of those works.

Kindly note that the previous works have been extensively addressed and revised in the "Introduction" section of the paper. Moreover, the relevant references have been added to the "References" section in accordance with these revisions.

3.     It should be clarified whether the proposed QSDNO is tested on a practical QKD network integrated into the current optical communication infrastructure. In that case, it is necessary to introduce the topology of the QKD network, including the scale of the optical communication infrastructure, the number of QKD devices, and so on.

Additionally, it should be noted that the attached paper has undergone significant revisions, resulting in clearer text, as well as the inclusion of additional figures and tables that present the results obtained from the live testbed. The primary contribution of this paper lies in the utilization of Software-Defined Networking (SDN) orchestration to integrate regionally separated networks. These networks employ distinct key management systems (KMS) that are managed by different SDN controllers. The aim is to ensure the provision of end-to-end Quantum Key Distribution (QKD) services, facilitating the delivery of QKD keys between geographically distinct QKD networks. Clause 4 of the paper outlines the actual data model and procedure deployed in the SDN orchestrator, providing modified texts and figures to elucidate the concepts.

4.     It is necessary to explain the difficulties of the communication between nodes from two different QKD networks. The specific method to overcome these difficulties should also be supplemented.

This has been clarified in the clause the last paragraph of clause 4.2.1 and clause 4.2.2 of the newly revised paper. Kindly also note our response to your question 3.

5.     The experimental results and analysis should be provided to help readers to evaluate the performance of the proposed scheme.

As mentioned in the response to question 3, the newly revised paper now includes a comprehensive description of the actual implementation and results obtained from the testbed. This updated version provides detailed insights into the implementation process and presents the outcomes and findings derived from the conducted experiments.

6.     The Section of CONCLUSION is missing.

n the revised paper, a dedicated section for the conclusion has been incorporated. This section encapsulates the key findings, implications, and overall significance of the research. By including a well-defined conclusion, the revised paper offers a comprehensive summary that solidifies the research's outcomes and brings the paper to a meaningful close.

  1. Figure 2 needs to be reorganized to make it more understandable.

Upon careful examination, the authors discovered that Figure 2 in the initial paper submission was deemed redundant and lacked clarity. As a result, this figure has been eliminated in the revised version of the paper. In its place, explanatory texts have been included within clause 4.2.2 to provide a clear understanding of the functions represented in Figure 6 of the revised paper.

As the use case described in the paper was practically implemented between 2020 and 2022 in the testbed in South Korea. It is now clearly indicated in clause 4 and the details of the implementation are given in particular in clause 4.2.

Please also note that the paper attached has been heavily revised with more clear text, more figures, and tables showing the results in the live testbed. The main contribution of this paper is the use of Software-Defined Networking (SDN) orchestration to integrate regionally separated networks where different network parts use incompatible key management systems (KMS) managed by different SDN controllers to ensure end-to-end QKD service provisioning to deliver the QKD keys between geographically different QKD networks. The real data model and the procedure deployed in the SDN orchestrator are also described in clause 4 with modified texts and figures.

The authors assert that the reviewer's comments have significantly contributed to the improvement of the paper's overall quality. The authors hope this revision clarifies the original intention and answers the questions raised by the reviewer.

Sincerely,

The authors of the aforementioned paper addressed in the heading of this cover letter

Reviewer 3 Report

The paper focuses on the seamless integration of QKD devices into the current  optical communication infrastructure, with the objective of managing individual SDN controllers for QKD equipment and transmission equipment and enabling the exchange of keys between different networks. This is an important and perspective topic. Since one of the foci of Enthropy is Information systems, it is also relevant to the scope of the journal.

I have some comments on the paper:

1. The authors claim that QKD "creates truly random keys...", but the main scope of this tecnhnology is key distribution. Quality of the keys depends on the source of enthropy and might be indeed better than "classica" if a quantum random number generation is used. I recommed the authors to alter the phrases a bit.

2. In the paragraph describing SDN I would suggest adding some references as well as clarifying that not every network device, i.e. router, can operate under and SDN controller, so upgrading the physical networks usually leads to additional costs (SDN may not be widely used, depending on the country)

3. Since this is not the first paper on QKD + SDN, I believe the authors should explicitly describe the novelty and add some literature, such as:

a. A. Aguado et al. “Secure NFV Orchestration over an SDN-Controlled Optical Network with Time-Shared Quantum Key Distribution Resources,” Journal of Lightwave Technology, vol. 35, no. 8, pp. 1357 – 1362, 2017.

b. V. R. Dasari et al. “Programmable Multi-Node Quantum Network Design and Simulation,” Proc. SPIE, Quantum Information and Computation, vol. IX, pp. 98730B, 2016.

c. Wang, Hua & Zhao, Yongli & Nag, Avishek. (2019). Quantum-Key-Distribution (QKD) Networks Enabled by Software-Defined Networks (SDN). Applied Sciences. 9. 2081. 10.3390/app9102081.

d. P. Comi et al., "Increasing network reliability by securing SDN communication with QKD," 2021 17th International Conference on the Design of Reliable Communication Networks (DRCN), Milano, Italy, 2021, pp. 1-3, doi: 10.1109/DRCN51631.2021.9477334.

etc.

A brief summary of 5-10 papers in the introduction would improve the quality of presentation and help stating the new results.

4. It if not entirely clear from the paper if the authors have created QSDN Orchestrator (QSDNO) or are just applying it to a particular network. What controller are they using: a free software or a proprietary one?

5. What QKD systems were deployed at the testbed? I might have missed it, but what software was running on the QKD kits and what on an external server? This is important for security assessment. 

Finally, the paper uses some inaccurate terms: "classical community", "traditional networking", which might be clarified.

Overall, the paper is well-written, but I am not sure what results are new, since there has been a lot of work on QKD + SND in the last 10 years, and a solid foundation has been laid. There are very few references and almost no summary of previous works nor a strong novelty statement.  Also the paper is a bit obscure on what type of physical equipment was used at KOREN testbed: orchestrator, colntroller, QKD kits, routers, etc. If the authors improve on these two issues, I will recommend the paper for publication.

Author Response

Dear Sir/Madam,

Thank you for your time and efforts to review the paper.

We have taken the time to individually address and respond to each of your comments and questions as follows:

  1. The authors claim that QKD "creates truly random keys...", but the main scope of this tecnhnology is key distribution. Quality of the keys depends on the source of enthropy and might be indeed better than "classica" if a quantum random number generation is used. I recommed the authors to alter the phrases a bit.

As recommended, the authors revised the texts to focus on the main contribution of the paper and modified texts accordingly.

  1. In the paragraph describing SDN I would suggest adding some references as well as clarifying that not every network device, i.e. router, can operate under and SDN controller, so upgrading the physical networks usually leads to additional costs (SDN may not be widely used, depending on the country)

Taking the recommendation into account, the authors have undertaken extensive revisions in the "Introduction" section. The revised version now emphasizes that Software-Defined Networking (SDN) holds promising potential for effectively managing QKD-enabled networks. It is important to note that this paper does not delve into classical network components such as routers. Instead, the authors specifically concentrate on the area addressed within the paper, thereby providing a focused examination of the subject matter.

  1. Since this is not the first paper on QKD + SDN, I believe the authors should explicitly describe the novelty and add some literature, such as:

a. A. Aguado et al. “Secure NFV Orchestration over an SDN-Controlled Optical Network with Time-Shared Quantum Key Distribution Resources,” Journal of Lightwave Technology, vol. 35, no. 8, pp. 1357 – 1362, 2017.

b. V. R. Dasari et al. “Programmable Multi-Node Quantum Network Design and Simulation,” Proc. SPIE, Quantum Information and Computation, vol. IX, pp. 98730B, 2016.

c. Wang, Hua & Zhao, Yongli & Nag, Avishek. (2019). Quantum-Key-Distribution (QKD) Networks Enabled by Software-Defined Networks (SDN). Applied Sciences. 9. 2081. 10.3390/app9102081.

d. P. Comi et al., "Increasing network reliability by securing SDN communication with QKD," 2021 17th International Conference on the Design of Reliable Communication Networks (DRCN), Milano, Italy, 2021, pp. 1-3, doi: 10.1109/DRCN51631.2021.9477334.

etc.

A brief summary of 5-10 papers in the introduction would improve the quality of presentation and help stating the new results.

Kindly note that the previous works have been extensively addressed and revised in the "Introduction" section of the paper. Moreover, the relevant references have been added to the "References" section in accordance with these revisions.

  1. It if not entirely clear from the paper if the authors have created QSDN Orchestrator (QSDNO) or are just applying it to a particular network. What controller are they using: a free software or a proprietary one?

The paper is based on the YANG model, which is widely adopted within the SDN community. However, it is worth mentioning that the controller and orchestrator used in this study are proprietary implementations developed by SK Telecom. These specific controller and orchestrator systems were chosen for the purposes of the research to evaluate their effectiveness in managing the QKD-enabled networks.

  1. What QKD systems were deployed at the testbed? I might have missed it, but what software was running on the QKD kits and what on an external server? This is important for security assessment.

Finally, the paper uses some inaccurate terms: "classical community", "traditional networking", which might be clarified.

Additionally, it should be noted that the attached paper has undergone significant revisions, resulting in clearer text, as well as the inclusion of additional figures and tables that present the results obtained from the live testbed. The primary contribution of this paper lies in the utilization of Software-Defined Networking (SDN) orchestration to integrate regionally separated networks. These networks employ distinct key management systems (KMS) that are managed by different SDN controllers. The aim is to ensure the provision of end-to-end Quantum Key Distribution (QKD) services, facilitating the delivery of QKD keys between geographically distinct QKD networks. Clause 4 of the paper outlines the actual data model and procedure deployed in the SDN orchestrator, providing modified texts and figures to elucidate the concepts.

Overall, the paper is well-written, but I am not sure what results are new, since there has been a lot of work on QKD + SND in the last 10 years, and a solid foundation has been laid. There are very few references and almost no summary of previous works nor a strong novelty statement.  Also the paper is a bit obscure on what type of physical equipment was used at KOREN testbed: orchestrator, colntroller, QKD kits, routers, etc. If the authors improve on these two issues, I will recommend the paper for publication.

As mentioned in the response to question 3 and 5, the newly revised paper now includes a comprehensive description of the actual implementation and results obtained from the testbed. This updated version provides detailed insights into the implementation process and presents the outcomes and findings derived from the conducted experiments

As the use case described in the paper was practically implemented between 2020 and 2022 in the testbed in South Korea, it is now clearly indicated in clause 4 and the details of the implementation are given in particular in clause 4.2.

Please also note that the paper attached has been heavily revised with more clear text, more figures, and tables showing the results in the live testbed. The main contribution of this paper is the use of Software-Defined Networking (SDN) orchestration to integrate regionally separated networks where different network parts use incompatible key management systems (KMS) managed by different SDN controllers to ensure end-to-end QKD service provisioning to deliver the QKD keys between geographically different QKD networks. The real data model and the procedure deployed in the SDN orchestrator are also described in clause 4 with modified texts and figures.

The authors assert that the reviewer's comments have significantly contributed to the improvement of the paper's overall quality. The authors hope this revision clarifies the original intention and answers the questions raised by the reviewer.

Sincerely,

The authors of the aforementioned paper addressed in the heading of this cover letter

Round 2

Reviewer 2 Report

The authors have carefully revised the manuscript according to my comments. I suggest to accept it to publish in ENTROPY after the following minor problems are solved.

1.       I suggest to delete Column 3 of Table 1 since the information may confuse the readers.

2.       The references to Table 1 in Line 450 and Line 467 are probably incorrect. Please check and correct them.

Reviewer 3 Report

The authors have made significant improvements of the paper based on my comments, I am satisfied with the revised version and recommend it for publication.